# Endothelial Autophagy in Coronary Microvascular Dysfunction and Cardiovascular Disease

**DOI:** 10.3390/cells11132081

**Published:** 2022-06-30

**Authors:** Fujie Zhao, Ganesh Satyanarayana, Zheng Zhang, Jianli Zhao, Xin-Liang Ma, Yajing Wang

**Affiliations:** 1Department of Emergency Medicine, Thomas Jefferson University, Philadelphia, PA 19107, USA; fujie.zhao@jefferson.edu (F.Z.); zhen.zhang@jefferson.edu (Z.Z.); jianli.zhao@jefferson.edu (J.Z.); xinliang.ma@jefferson.edu (X.-L.M.); 2Emory Eye Center, Emory University School of Medicine, Atlanta, GA 30322, USA; ganesh.satyanarayana@emory.edu

**Keywords:** coronary microvascular dysfunction, endothelial cell, autophagy, cardiovascular disease

## Abstract

Coronary microvascular dysfunction (CMD) refers to a subset of structural and/or functional disorders of coronary microcirculation that lead to impaired coronary blood flow and eventually myocardial ischemia. Amid the growing knowledge of the pathophysiological mechanisms and the development of advanced tools for assessment, CMD has emerged as a prevalent cause of a broad spectrum of cardiovascular diseases (CVDs), including obstructive and nonobstructive coronary artery disease, diabetic cardiomyopathy, and heart failure with preserved ejection fraction. Of note, the endothelium exerts vital functions in regulating coronary microvascular and cardiac function. Importantly, insufficient or uncontrolled activation of endothelial autophagy facilitates the pathogenesis of CMD in diverse CVDs. Here, we review the progress in understanding the pathophysiological mechanisms of autophagy in coronary endothelial cells and discuss their potential role in CMD and CVDs.

## 1. Introduction

The term “coronary microvascular dysfunction” (CMD) refers to a subset of structural and/or functional disorders of coronary microcirculation that lead to impaired coronary blood flow (CBF) and ultimately result in myocardial ischemia [1,2,3] (Figure 1B). With recent advances in the knowledge of the pathophysiological mechanisms and the development of more sophisticated tools for assessment, CMD has emerged as a prevalent cause of microvascular angina (MVA), a condition where patients present angina and myocardial ischemia but without evidence of obstructive coronary artery disease (CAD). CMD has also been linked to multiple other diseases, such as obstructive CAD, diabetic cardiomyopathy (DCM), primary cardiomyopathies, myocarditis, and heart failure (HF), particularly heart failure with preserved ejection fraction (HFpEF) [4] (Figure 1C). Even subjects without clinical manifestations of heart diseases but with classic risk factors (e.g., smoking, hypercholesterolemia, hypertension, and obesity) or chronic inflammatory vessel disease present CMD [5,6,7]. Autophagy is a critical regulator of cardiac metabolism and homeostasis [8,9,10]. Deficient or uncontrolled activation of endothelial autophagy is associated with the onset and development of diverse cardiovascular diseases (CVDs), including CMD [9,10,11,12,13,14,15,16]. Understanding the molecular basis of endothelial autophagy in CMD is crucial for the discovery of novel regulatory mechanisms and the identification of new diagnostic and therapeutic targets. This review aims to summarize the pathophysiological mechanisms of autophagy in coronary endothelial cells (ECs) and explore their potential role in CMD and CVDs.

## 2. The Coronary Microvasculature and Endothelium in the Heart

The coronary arterial system is a continuous network with a decreasing size and distinct functions [1,3,17,18], which dynamically delivers oxygen, nutrients, and hormones to the myocardium and removes metabolic end-products [19,20]. The human coronary vasculature consists of the proximal large epicardial coronary arteries (>400 µm), small pre-arterioles (100 to 400 µm), smaller intramural arterioles (<100 µm), and the coronary capillary bed (<10 µm) [1,18] (Figure 1A). The proximal large epicardial coronaries serve as conductance vessels and present limited resistance to CBF; they change their diameters with shear stress and endothelial function [21]. Conversely, the pre-arterioles and arterioles exhibit remarkable resistance to CBF and maintain the pressure within a narrow range along with the perfusion pressure or flow variations [3,17,18]. Smaller intramural arterioles function to match the myocardial blood supply with the oxygen consumption [20,22] and contribute to the largest proportion of the entire coronary vascular resistance. Pre-arterioles, intramural arterioles, and capillaries make up the coronary microcirculation.

The subset of disorders including structural and/or functional abnormalities in the coronary microcirculation leading to an impaired coronary blood supply is called CMD (Figure 1B). Examples of structural abnormalities and microvascular remodeling in CMD include intramural arteriole stenosis, perivascular fibrosis, and capillary rarefaction (often in the context of increased left ventricular mass) [3], particularly in patients with CAD risk factors or underlying cardiomyopathies. Of note, the functional abnormalities present as impaired dilations (vasodilator abnormalities) and/or increased constriction of the coronary microvessels (i.e., microvascular spasms) [1] are mediated by EC-dependent and/or EC-independent mechanisms [2,3,4,23]. In particular, the vascular ECs play a pivotal role in modulating vasomotor activity by releasing vasoactive substances [24]. Important EC-derived vasodilators are nitric oxide (NO), prostacyclin, bradykinin, and the EC-derived hyperpolarizing factor (EDHF) [24,25,26,27,28,29]. The EC-derived vasoconstrictors include endothelin-1 (ET-1), prostaglandin H2, thromboxane A2, and superoxide anions [30,31]. Moreover, EC-derived vasodilators, such as nitric oxide, could retard cellular growth/migration and exhibit potent antiatherogenic/thromboresistant properties by inhibiting platelet aggregation and cell adhesion [23], whereas EC-derived vasoconstrictors display counterbalanced effects within a given vascular segment [23].

Endothelial dysfunction in resistant coronary vessels has proved to be an essential contributor to CMD [32,33,34]. Endothelial dysfunction is identified as a disturbed vasodilatory or hyperactivated vasoconstrictive reaction to the EC-dependent vasodilator Ach [32,33] along with augmented oxidative stress, increased reactive oxygen species (ROS) and/or vasoconstrictors production, and gradually decreased endothelial NO availability. Furthermore, endothelial dysfunction also includes the transformation from a quiescent state to an activated pro-inflammatory state, causing promoted chemokine and adhesion molecule expression and enhanced consecutive interaction with platelets and leukocytes [35,36,37], which ultimately results in microvascular structural remodeling and contributes to myocardial perfusion defects.

## 3. Brief Overview of Autophagy

Autophagy is a highly conserved process in which obsolete and dysfunctional cytoplasmic components (such as unfolded proteins, lipids, and damaged organelles) are degraded and recycled, and infectious organisms are removed by lysosomes [10,11,38,39]. Autophagy is stimulated by different stresses (i.e., nutrient deprivation and hypoxia) and functions primarily as a cell survival mechanism [40,41,42,43]. However, it switches to promoting cell death under insurmountable lethal stress, which is known as autophagic (type II) cell death [44,45]. Three classes of autophagy are identified as follows: chaperone-required autophagy, microautophagy, and macroautophagy [46,47,48]. Chaperone-required autophagy is the selective degradation of proteins with a KFERQ-like motif. During the process, targeted proteins are transferred to the lysosomes with the company of the chaperone HSC70 and co-chaperones, subsequently internalized into the lysosomes via an interaction with the lysosome-associated membrane protein type 2A (LAMP2A) [49]. In microautophagy, the cargo, alone or in a complex with chaperones, can be directly engulfed by the lysosome and late endosomes through invagination at the lysosomal membrane via electrostatic forces [50,51,52]. Macroautophagy (hereafter referred to as autophagy) is the most well studied and the major type of autophagy. The macroautophagy pathway is characterized by the formation of autophagosomes, within which cytoplasmic components are insulated and subsequently degraded by fusing with the lysosomes [12,45,53,54]. The process of autophagy can be dissected into the following sequential steps: induction of a phagophore assembly site, nucleation of an autophagosome precursor (known as the phagophore), membrane expansion and maturation of the autophagosome (a double membrane vesicle), fusion with the lysosome for degradation, and lastly, recycling the degraded cargo [10,11,55,56].

The process of autophagy is sequentially regulated by multiple mechanisms (Figure 2). (1) First, initiation of the phagophore assembly begins with the formation of a preinitiation complex, which is composed of the unc-51-like kinase (ULK1/2), autophagy-related protein 13 (ATG13), and the non-catalytic focal adhesion kinase-family interacting protein of 200 kD (FIP200) [45,57,58]. The activity of this kinase complex is negatively inhibited by the mammalian target of the rapamycin complex 1 (mTORC1) and the positively activated AMP-activated protein kinase (AMPK) [59,60]. (2) Further nucleation involves the recruitment and activation of the initiation complex, which is composed of the vacuolar protein sorting protein 15 (VPS15), a class III PI3K (VPS34), and Beclin 1 [61,62]. The activity of the initiation complex is downregulated by several independent signaling pathways, such as the PI3K-AKT pathway, and the Bcl2 and Bcl-xL pathway [63,64]. Conversely, starvation or exercise-induced autophagy activation is carried out by the JNK family [65,66]. (3) Next, the phagophore elongation and closure require two distinct but complementary ubiquitin-like protein conjugation systems—the ATG12/ATG5/ATG16L1 complex and the microtubule-associated protein 1-light chain 3 (LC3)-phosphatidylethanolamine (PE) machinery [45]. (4) Finally, the fusion of the autophagosomes with the lysosomes requires the syntaxin17 (Stx17) synaptosome-associated protein 29 (SNAP29), the vesicle-associated membrane protein 8 (VAMP8), and the lysosomal-associated membrane protein 1/2 (LAMP1/2) [67,68,69,70].

The selective sequestration of the dysfunctional mitochondria by the autophagosomes and the subsequent degradation by the lysosomes are termed "mitophagy” [12] (Figure 3). There are two distinct signaling pathways for mitophagy, which are as follows: the phosphatase and tensin homolog (PTEN)-induced putative kinase 1 (PINK1)–Parkin-dependent pathway and the mitophagy receptor-required pathway [71,72]. To date, multiple mitophagy receptors have been recognized in mammals, including the BCL2/adenovirus E1B 19kDa-interacting protein 3 (BNIP3) [73], NIP3-like protein X (NIX, also called the BNIP3-like protein (BNIP3L)) [74,75,76], FUN14 domain-containing protein 1 (FUNDC1) [77], B-cell lymphoma-2-like 13 (BCL2L13) [78], and FK506-binding protein 8 (FKBP8) [79].

## 4. Coronary Endothelial Autophagy in CVDs

In the past, autophagy (including mitophagy) in cardiomyocytes was thought to play a predominant role in heart injuries. With the increasing recognition of the crucial contributions of CMD to CVDs, the role of autophagy (including mitophagy) in non-myocytes, particularly in coronary ECs, has attracted great interest. Emerging evidence unravels that autophagy is required for multiple EC functions, such as secretion of adhesion molecules [80,81], EC nitric oxide synthase (eNOS)-derived NO bioavailability [82,83,84], expression of ET-1 [85,86], ROS production [83,87], and inflammatory cytokines production [83], which participate in a wide range of cellular events, including endothelial proliferation [86], senescence [87,88,89], and apoptosis [88,90,91,92]. EC-intrinsic autophagy is suggested to enable ECs to adjust plastically to various insulting stressors [85,90,93] and leads to autophagic cell death in severely damaged ECs [94,95,96]. Indeed, various studies have suggested that the EC-autophagic flux facilitates the pathogenesis of CMD in diverse CVDs, including ischemic heart disease, DCM, hypertrophy cardiomyopathy, heart failure, and inflammatory disorders [9,10,12,13,14,15,16]. However, the detailed role of autophagy in adjusting EC reactions is still controversial and seems to depend on the particular type of metabolic insults or the disease-associated microenvironment. Here, we will discuss the specific role of coronary endothelial autophagy (including mitophagy) in CVDs in detail (summarized in Appendix A).

### 4.1. Coronary Endothelial Autophagy in Obstructive CAD (Stable CAD and Acute Coronary Syndromes)

In the hearts of obstructive CAD patients, CMD probably coexists and plays a role in causing myocardial ischemia in regions perfused by arteries both with and without stenosis. Thus, CMD has important diagnostic, prognostic, and management implications [1,18,97,98]. In regions distal to arterial stenosis, the chronic modulation of coronary microcirculation to limited perfusion pressure may negatively impact the microvascular remodeling and the maximal capacity of vasodilation after restoring to normal CBF [99]. The most severe form of CMD is microvascular obstruction (MVO), which refers to a capillary destruction with a no-reflow phenomenon, despite recanalization of the epicardial coronary artery [100,101,102]. The pathogenic mechanisms underlying CMD in obstructive CAD include coronary microvascular EC injuries, which may occur much earlier and with much more severe damage than cardiomyocyte injuries. Emerging evidence shows that autophagy plays a fundamental role in this process [103,104]. However, the underlying mechanisms remain controversial. Upon oxidative stress, Wang et al. reported that microRNA-103 could protect the human coronary artery ECs (HCAECs) against H_2_O_2_-induced injuries by preventing the Bcl-2/BNIP3-mediated suppression of end-stage autophagy [105]. Furthermore, by using mice with EC-specific NADPH oxidase 2 (Nox2)/gp91 overexpression, Shafique et al. demonstrated that endogenous ROS oxidative stress protected mouse heart ECs from oxidant-induced cell death by increasing the AMPK-mTOR-mediated autophagy and through improving the AMPK-eNOS-mediated EC-dependent-coronary vasodilatation [106]. Autophagy can also be induced by hypoxia stress [107]. Wang et al. identified that forkhead box O3 alpha (foxO3α)-dependent autophagy aggravated hypoxia-induced rat cardiac microvascular endothelial cell (CMEC) dysfunction and apoptosis [108]. However, according to Sun’s study, the mitophagy induced by the Rcan1-1L (regulator of calcineurin 1-1L) overexpression contributed to cell survival under hypoxic conditions [109]. During an oxygen–glucose deprivation and reoxygenation (OGD/R) injury, Shao et al. reported that dexmedetomidine could protect the CMECs by activating the peroxisome proliferator-activated receptors (PPARδ)-AMPK-PGC-1α pathway-dependent autophagy, effectively causing a decrease in ROS production and an increase in cell viability [110]. Nevertheless, inhibited autophagy participated in the beneficial role of polysaccharides from Enteromorpha Prolifera (PEP) in OGD-induced human CMEC injury by promoting the mTOR pathway [111].

Notably, the role of endothelial autophagy in hypoxia/reoxygenation (H/R) or ischemia/reperfusion (I/R)-induced injury is much more complex. Generally, autophagy is a protective mechanism against cardiac H/R or I/R injury [112]. Its protective role is driven by inducing the mitogen-activated protein kinase (MEK)/extracellular signal-regulated kinase (ERK) pathway [113] or via activating and promoting the transcription factor EB (TFEB) translocating from the lysosomes to the nuclei [114]. Furthermore, mitophagy was usually referred to as a pro-survival regulator to I/R or H/R injury [115]. Inhibition of the FUNDC1-mediated mitophagy in CMECs by the nuclear receptor subfamily 4 group A member 1 (NR4A1) [116] or receptor-interacting protein kinase 3 (Ripk3) [117] exhibited the disturbed mitochondrial homeostasis, upregulated the expression of EC-derived pro-inflammatory and adhesive factors, enhanced endothelial apoptosis, and provoked CMD in cardiac I/R injuries. In addition, enhancing autophagy through Beclin1 overexpression in CMECs exhibited suppressed NACHT, LRR, and PYD domain-containing protein 3 (NLRP3) inflammasome activation by promoting tumor necrosis factor-alpha-induced protein 3 (TNFAIP3) [118] and inhibited caspase-4 inflammasome activation [119]. Thus, it resulted in a reduced IL-1β level and an increased animal survival upon myocardial I/R injuries. In addition, the sarcoplasmic/endoplasmic reticulum Ca2+-ATPase (SERCA) overexpression [120] and miR-92a-3p inhibition [121] could protect the CMECs against myocardial I/R injuries by preserving the EC mitophagy. However, hyperautophagy is linked to I/R or H/R injury-induced mitochondria and EC apoptosis. The inhibition of autophagy contributed to the anti-apoptosis effects of glycyrrhizic acid (GA) in H/R-induced CMEC injury [122]. Melatonin was reported to play a beneficial role in CMECs against I/R injury through directly suppressing autophagy via the AMPK/mTOR pathway [123] or by inhibiting the mitophagy-mediated cell death via the dynamin-related protein 1 (Drp1)-voltage-dependent anion channel 1 (VDAC1)-hexokinase 2 (HK2)-mitochondrial permeability transition pore (mPTP)-PINK1/Parkin axis in an AMPKα-dependent manner [124]. Furthermore, neuregulin-1β (Nrg1β) protected the cardiac ECs against I/R injury by preventing ATG5-required autophagy-induced Trx2 (thioredoxin) degradation and rescuing eNOS function via upregulating the Erb-B2 receptor tyrosine kinase 2 (ErbB2) [125]. In addition, Diao et al. found that a long noncoding RNA (LncRNA) UCA1 transferred from human umbilical cord mesenchymal stem cell-derived exosomes (hUCMSC-ex) protected the CMECs against H/R injury by inhibiting autophagy via damaging the miR-143-mediated degradation of Bcl-2 [126]. Importantly, ischemia and I/R injury also evoke a dramatic autophagic flux in cardiomyocytes, which could either serve as a pro-survival mechanism to meet metabolic demands and eliminate damaged cellular components and organelles, or function as a pro-death mechanism to initiate apoptosis. The crosstalk between endothelial and cardiac autophagy is of great interest but poorly understood. Further investigation is urgently needed.

#### Autophagy in Percutaneous Coronary Intervention (PCI)-Associated Coronary EC Injury

Accumulative evidence suggests that coronary endothelial autophagy associated with drug-eluting stents (DESs) underlies PCI-associated complications, such as early/late stent thrombosis, MI, in-stent restenosis (ISR), and mortality [127,128,129]. Shin-ichiro et al. showed that endothelial autophagy differentially induced by sirolimus and paclitaxel via regulating LC3B, Bcl-2, and p53 was one of the potential mechanisms in suppressing re-endothelialization and revascularization, as well as NO production in human aortic ECs [130]. In addition, epigallocatechin-3-gallate (EGCG) and nickel-containing austenitic 316L stainless steel (316L SS) were reported to inhibit human umbilical vein EC (HUVECs) proliferation by upregulating the autophagic genes ATG 5/7/12 and the cell apoptosis genes caspase 3/8/9 and Fas to suppress the occurrence of ISR [131,132]. Moreover, autophagy-dependent endothelial membrane remodeling is crucial for rapamycin-eluting stent-induced stent thrombosis [133]. In sum, autophagy-targeted drugs may serve as promising preventive targets for PCI-associated complications.

### 4.2. Coronary Endothelial Autophagy in HFpEF

HFpEF is recognized as a clinically heterogeneous syndrome, in which patients present classic symptoms and signs of HF but exhibit a normal or near-normal EF [134,135]. CMD is hypothesized to play a fundamental role in the pathophysiological process of HFpEF [136,137,138,139]. Up to 75% of HFpEF patients exhibit impaired CFR in spite of the absence of obstructive CAD [140]. Diabetes mellitus (DM), metabolic syndrome, hypertension, and obesity are prevalent cardiovascular risk factors for HFpEF [134,136,141]; they aggravate cardiac dysfunction and remodeling through CMD [142]. The basic mechanisms that mediate the progression of HFpEF include myocyte hypertrophy, energetic imbalance, mitochondrial dysfunction, EC dysfunction, increased oxidative stress, inflammation, interstitial fibrosis, and damaged angiogenesis [143,144,145,146]. The pathophysiological role of autophagy in HFpEF onset and progression has just begun to be acknowledged. Here, we mainly focus on the impact of autophagy on cardiac angiogenesis and fibrosis, two critical events associated with the progression of HFpEF.

#### 4.2.1. Coronary Endothelial Autophagy in Angiogenesis

Coronary microvascular rarefaction is consistently linked to most heart diseases, particularly in hypertrophic cardiomyopathy and HFpEF [137,142]. Angiogenesis, known as the development of new blood vessels and vascular circuits from pre-existing vessels [147,148], is a potent mechanism to reverse rarefaction. Autophagy is suggested to promote angiogenesis and play a crucial role in maintaining vessel wall homeostasis [149,150,151,152,153,154]. In a mouse model of acute myocardial infarction (AMI), VEGF-A proved to improve angiogenesis after AMI by promoting ROS production and increasing ER stress-induced autophagy [151]. In addition, AGGF1 (angiogenic factor with G-patch and FHA domain 1) was recognized as an autophagy initiation inducer in ECs through the activation of JNK, which resulted in Vps34 lipid kinase activation and increased Beclin1-Vps34-ATG14 complex assembly [93]. In the MI model, AGGF1 KO mice showed downregulated autophagy and inhibited angiogenesis with larger infarct areas and severe contractile dysfunction [93]. Furthermore, in a mouse model of TAC, an EC leptin receptor deletion was shown to promote cardiac angiogenesis by enhancing autophagosome formation by suppressing AKT/mTOR signaling, leading to reduced cardiac inflammation and fibrosis and improved left ventricular function [152]. In addition, a sirtuin 3 (Sirt3) deletion was found to aggravate angiotensin II-induced PINK1/Parkin acetylation aberrance, resulting in impaired mitophagy, excessive mtROS generation, and damaged angiogenic capacity of primary mouse CMECs [153]. Moreover, reduced trichoplein (TCHP, a centriolar satellites protein) and high p62 levels were detected in primary ECs from patients with CAD [154]. The TCHP knockout (KO) mice had impaired autophagosome maturation, accumulated p62 in the heart and cardiac vessels, and damaged cardiac vascularization [154]. Of note, TFEB was reported as a positive regulator of angiogenesis through the activation of AMPKα and autophagy via the TFEB-dependent transcriptional upregulation of MCOLN1 (mucolipin-1) in the mouse hindlimb ischemia model [155], which implies its potential cardioprotective role in ischemic heart disease. However, some opposite evidence supports the anti-angiogenic effect of autophagy in certain situations. Liu et al. reported that MGO (methylglyoxal), a metabolite of glycolysis, which was upregulated in diabetic patients, exhibited reduced endothelial angiogenesis through the receptor for advanced glycation end products (RAGE)-required, peroxynitrite (ONOO)-mediated, and autophagy-triggered VEGFR2 degradation, which might represent a novel mechanism for DM-associated angiogenesis abnormalities [156]. Furthermore, soluble RAGE exerted a cardioprotective effect through the inhibition of autophagy by activating the signal transducer and activator of the transcription 3 (STAT3) pathway, causing increased angiogenesis and reduced cardiomyocyte apoptosis in I/R-injured mice and OGD/R-insulted CMECs [157,158]. In addition, Pan et al. showed that capsicodendrin (CPCD), a natural compound isolated from Cinnamosma Macrocarpa, inhibited sprouting angiogenesis during zebrafish embryonic development by enhancing autophagy in ECs via inhibiting the VEGFR2/AKT pathway [159]. Moreover, silica nanoparticles (SiNPs), one of the most widely applied engineered nanomaterials, could enhance the autophagic activity, disturb EC homeostasis, and impair angiogenesis (via an inhibitory effect on ICAM-1 and VCAM-1 expression) [160]. To date, besides endovascular intervention, limited therapeutic options to improve CBF in ischemic hearts are available. Therefore, EC autophagy, as a significant regulator of angiogenesis, would be a promising therapeutic target to promote CBF.

#### 4.2.2. The Role of Autophagy in Endothelial–Mesenchymal Transition (EndMT)-Mediated Cardiac Fibrosis

Cardiac fibrosis is a primary consequence of CMD and contributes to nearly all forms of heart disease. The persistence of fibrosis results in progressive ventricular wall stiffening, reduced contractility, and cardiac conductance abnormalities, ultimately leading to heart failure and cardiac death [161,162,163]. Although activated fibroblasts, named myofibroblasts, are regarded as the predominant regulator for fibrosis, the contribution of EndMT to the initiation and progression of fibrosis has been gradually appreciated [164,165]. Recent research suggests that the inhibition of autophagy to EndMT serves as a critical cardioprotective mechanism in ameliorating cardiac fibrosis [149,150,166,167,168]. According to Ke’s study, restoring the TFEB-mediated autophagic flux could inhibit the transforming growth factor-β (TGF-β)-meditated EndMT and promote angiogenesis in HCAECs by triggering TFEB nucleus translocation [166]. Of importance, Takagaki et al. showed that the disruption of EC autophagy (through an EC-specific ATG5 deletion) evidently induced pathological interleukin-6 (IL6)-dependent EndMT and aggravated heart fibrosis [167]. In addition, the upregulation of autophagy was reported to prevent hypoxia-induced EndMT and cell apoptosis, while enhancing angiogenesis by inhibiting the NF-κB-Snail signaling pathway in human CMECs [149,150]. Furthermore, Pan et al. identified that irisin (a new hormone-like myokine, primarily secreted by cardiomyocytes) treatment significantly alleviated doxorubicin-initiated cardiac perivascular fibrosis by restraining EndMT via restoring autophagy in ECs, resulting in reduced ROS accumulation and inhibited NF-κB-Snail pathway [168]. However, some researchers found that in certain conditions, the activation of autophagy could induce EndMT and contribute to cardiac fibrosis [169,170]. Zhang’s study implied that the excessive activation of autophagy contributed to transverse aortic constriction (TAC)-induced cardiac fibrosis through upregulating EndMT, whereas the suppression of autophagy by inactivating RAGE partly reversed this phenomenon [169]. Moreover, Sasaki et al. showed that inducing autophagy by rapamycin promoted H2O2-induced EndMT through activating the TGF-β pathway [170]. Though more mechanistic studies are needed, autophagy-mediated EndMT still emerges as a promising therapeutic target to prevent the development of cardiac fibrosis.

### 4.3. Coronary Endothelial Autophagy in DCM

Clinical studies have shown that CMD is an early feature of DCM (even in patients without obstructive CVDs) [171,172,173] and this impairment is more pronounced in type 2 DM patients [174,175,176,177]. DCM studies in both animals and humans have emphasized the substantial role of the coronary ECs, particularly in the early stages of damage, in promoting ROS generation, and facilitating the recruitment of inflammatory cells. This ultimately resulted in myocardial microvascular rarefaction, diminished angiogenesis, and HFpEF phenotype [178,179,180]. Insulin signaling impairment, hyperglycemia/glucotoxicity, and lipotoxicity are predominant pathophysiological causes of DM-related CMD. Dysregulated autophagy is a key underlying cause in the onset and progression of DCM. A prolonged exposure of a fetal mouse heart to sugars (sucrose or mannitol) could induce severe lysosomal derangements and prominent autophagy in the ECs [181]. Mst1 (mammalian sterile 20-like kinase 1) is a serine/threonine kinase that functions as a negative regulator of autophagy in the heart by enhancing the binding of Beclin1 to Bcl-2 and promoting apoptosis by releasing Bcl-2 from Bax [182,183]. Hu et al. showed that Mst1-enriched exosomes excreted by CMECs were taken up by cardiomyocytes, resulting in inhibited autophagy and ultimately exacerbated high glucose (HG)-induced apoptosis in cardiomyocytes [184]. Meanwhile, Mst1 directly participated in the pathogenesis of CMD by inhibiting autophagy and increasing apoptosis in CMECs [185]. Furthermore, the upregulation of autophagy was reported to rescue HG-induced EC apoptosis through the AKT-mTOR signal pathway [186]. In addition, mitophagy was shown to protect mitochondrial integrity and prevent HG and palmitate acid (HG/PA)-induced EC apoptosis via the PINK1–Parkin pathway [187,188] and hinder HG/PA-induced EC senescence via the AMPK pathway [189]. Furthermore, improving Bnip3-dependent mitophagy could rescue ox-LDL-induced EC damage, resulting in a restored mitochondrial respiration complex activation, reduced ROS production, and an increased EC viability [190]. Interestingly, in certain conditions, the inhibition of autophagy can be protective. The downregulation of autophagy was reported to relieve HG-induced endothelial impairment via the glioma-associated oncogene homolog 1 (GLI1)-dependent-Hedgehog pathway [191].

Notably, endothelial autophagy also contributes to the shift of the myocardial metabolome [192]. In Altamimi’s study, the downregulation of EC autophagy by ATG7 KO impaired cardiac fatty acid stores and repressed the reliance of the heart on fatty acid oxidation as the primary fuel source both upon insulin insult and during reperfusion of cardiac ischemia [193]. Of interest, TNF-α-induced endothelial autophagy cooperated with the NF-κB signaling and resulted in upregulated fatty acid transporter protein 4 (FATP4) expression in CMECs, which finally facilitated CMEC PA transcytosis and aggravated insulin resistance [194].

### 4.4. Coronary Endothelial Autophagy in Other Heart Diseases

Except for the diseases mentioned above, coronary autophagy was also reported to participate in many other diseases. Kawasaki’s disease (KD) is a systemic febrile vasculitis and can lead to abnormalities of the coronary artery in about 25% of untreated cases, which has been reported as the predominant cause of children’s acquired heart diseases [195,196]. According to Qin’s report, the peripheral blood mononuclear cells (PBMCs) collected from KD patients with fever could induce autophagy in HCAECs, thus, promoting the secretion of chemokines and pro-inflammatory factors [197]. Moreover, ginsenoside Rb1 could effectively alleviate coronary artery lesions in a mouse KD model, possibly by upregulating the AMPK/mTOR/P70S6 pathway-mediated autophagy to prevent EC injury [198]. Additionally, the activation of autophagy was also involved in the anti-inflammatory effects of resveratrol in TNF-α-treated HCAECs [199]. Furthermore, autophagy was found to be upregulated during zebrafish heart regeneration and was positively correlated with the metformin-mediated cardiac regeneration acceleration in zebrafish, including epicardial, endocardial, and vascular endothelial regeneration [200]. Moreover, recent studies show that in aged EC compartments, autophagic activities are compromised [89]. Accordingly, in comparison with ECs in younger mice, ECs from older mice displayed lower levels of vital proautophagic proteins, such as Beclin1 and LC3 [201].

Notably, after coronary angiography, up to 40% of patients with typical clinical manifestations of myocardial ischemia were found with normal or near-normal appearing coronary arteries [202,203,204,205,206,207]. This situation is termed “MVA”, in which CMD is the principal alteration causing symptoms [202]. In particular, MVA is the disease that fosters the concept of CMD and draws people’s attention to the role of CMD in nonobstructive heart diseases for the first time. Unfortunately, to date, there is no animal model for MVA. Mechanistic research is urgently needed. Given the critical role of autophagy in CMD and diseases noted above, autophagy is believed to be a promising field for basic research in MVA.

## 5. Conclusions

CMD encompasses several pathogenetic mechanisms involving structural and functional impairments of the coronary microcirculation. It plays a pathophysiological and prognostic role among a broad range of CVDs and associated risk factors. However, its structural, functional, and molecular mechanisms have not been well clarified. To date, there is no specific CMD-targeted therapeutic intervention validated by large-scale randomized clinical trials. Herein, we provided a contemporary review that summarized the experimental evidence for the substantial modulatory role of coronary EC autophagy in CMD and various CVDs, which could be beneficial to basic and clinically oriented studies and could facilitate the innovation of novel diagnostic strategies for CMD-associated diseases. Of note, mitophagy is pivotal for mitochondrial quality control and plays a dual role in the progression of diverse CVDs. The emerging role of alternative forms of mitophagy in CVDs is well summarized in Pedro, Morales, and Li’s studies [13,208,209]. However, since the current studies are mainly focused on the role of mitophagy in cardiomyocytes, vascular smooth muscle cells, and ECs in the aorta, their roles in coronary ECs are less mentioned and further studies are warranted.

In addition, autophagy-targeted pharmacological and nutritional interventions, including mTORC1 inhibitors, AMPK activators, caloric restriction, caloric restriction mimetics, natural compounds, and specific miRNAs, are emerging as potential therapeutic candidates in patients with CVDs [9]. Given that CMD has emerged as a crucial denominator in diverse CVDs, additional bedside-to-bench studies in this field, particularly in coronary EC autophagy, are urgently needed.

## Figures and Tables

**Figure 1 cells-11-02081-f001:**
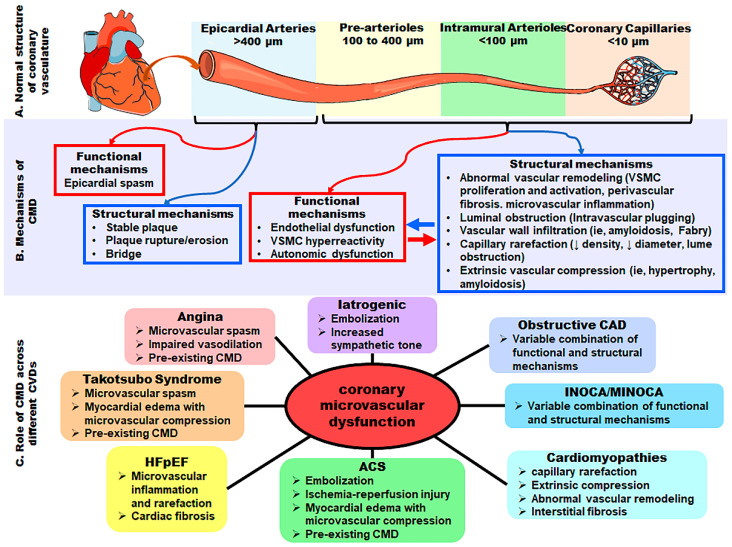
The coronary microvasculature in the heart. (**A**) Normal structure of the coronary vasculature. (**B**) Mechanisms of coronary microvascular dysfunction (CMD). (**C**) Role of CMD across different cardiovascular diseases (CVDs). Abbreviations: VSMCs, vascular smooth muscle cells; HFpEF, heart failure with preserved ejection fraction; ACS, acute coronary syndrome; INOCA, ischemia with nonobstructive coronary arteries; MINOCA, myocardial infarction with nonobstructive coronary arteries; CAD, coronary artery disease.

**Figure 2 cells-11-02081-f002:**
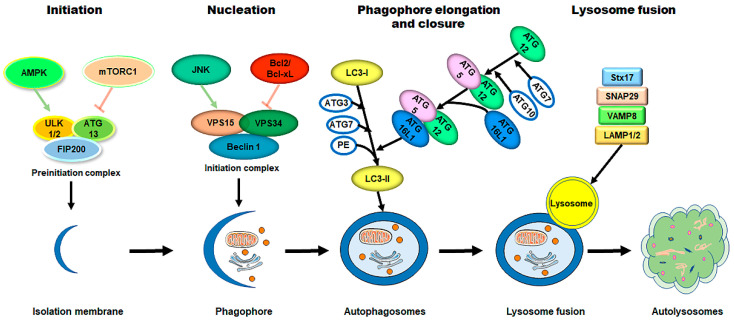
The process of autophagy in mammalian cells. The process of autophagy is sequentially dissected into several phases, including initiation, nucleation, phagophore elongation/closure, and autophagosome–lysosome fusion. (1) First, upon metabolic insults, the AMPK activation and/or mTORC1 inhibition result in the initiation of the preinitiation complex (ULK1/2, ATG13, and FIP200). (2) Further nucleation involves the recruiting and activating of the initiation complex (VPS15, VPS34, and Beclin 1), which is downregulated by the Bcl2/Bcl-xL pathways and upregulated by the JNK family. (3) Next, phagophore elongation and closure require the ATG12/ATG5/ATG16L1 complex and the LC3-PE machinery. (4) Finally, the fusion of the autophagosomes with the lysosomes requires Stx17, SNAP29, VAMP8, and LAMP1/2. Abbreviations: AMPK, AMP-activated protein kinase; mTORC1, mammalian target of rapamycin complex 1; ULK1/2, unc-51-like kinase 1/2; ATG, autophagy-related protein; FIP200, the non-catalytic focal adhesion kinase-family interacting protein of 200 kD; VPS, vacuolar protein sorting; LC3, microtubule-associated protein 1-light chain 3; PE, phosphatidylethanolamine; Stx17, syntaxin 17; SNAP29, synaptosome-associated protein 29; VAMP8, vesicle-associated membrane protein 8; LAMP1/2, lysosomal-associated membrane protein 1/2.

**Figure 3 cells-11-02081-f003:**
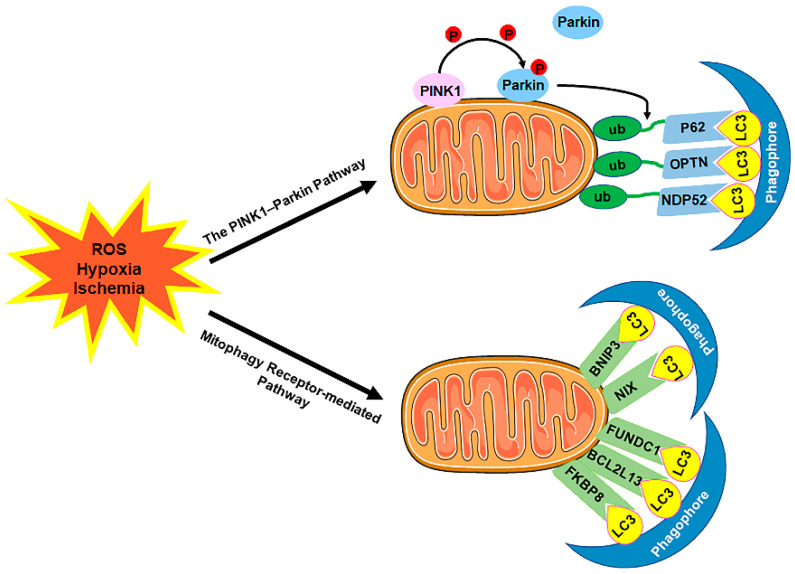
The mechanism of mitophagy. Mitophagy is induced by ROS, hypoxia, ischemia, and other stimuli. There are two distinct signaling pathways for mitophagy, which are as follows: (1) The PINK1–Parkin pathway of mitophagy. Following stress, PINK1 accumulates on the outer membrane of the mitochondria (OMM), promoting Parkin recruitment to ubiquitinate several OMM components. Poly-Ub chains are subsequently recognized by adaptor proteins (p62, OPTN, and NDP52) and further initiate autophagosome formations through binding with LC3. (2) The mitophagy receptor-mediated pathway. The BNIP3, NIX, FUNDC1, BCL2L13, and FKBP8 mitophagy receptors localize to the OMM and directly bind with LC3 to mediate mitochondrial elimination. Abbreviations: PINK1, PTEN-induced putative kinase 1; Parkin, Parkin RBR E3 ubiquitin-protein ligase; ub, ubiquitination; p62/SQSTM1, sequestosome 1; OPTN, optineurin; NDP52/CALCOCO2, calcium binding and coiled-coil domain 2; LC3, microtubule-associated protein 1A/1B-light chain 3; BNIP3, BCL2/adenovirus E1B 19 kDa-interacting protein 3; NIX, NIP3-like protein X; FUNDC1, FUN14 domain-containing 1; BCL2L13, B-cell lymphoma-2-like 13; FKBP8, FK506-binding protein 8.

## Data Availability

Not applicable.

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
