# Peer review of "Endothelial Autophagy in Coronary Microvascular Dysfunction and Cardiovascular Disease"

_cells, 2022, doi:10.3390/cells11132081_

Round 1

Reviewer 1 Report

The authors reviewed current literature about the role of endothelial autophagy in coronoray microvascular dysfunction and cardiovascular diseases. The review is well structured and written. I have some minor suggestions:

*Page 4. The authors should also discuss the emerging role of alternative forms of mitophagy in cardiovascular diseases.

*I understand that the focus of the review is to discuss the role of endothelial autophagy. However, I suggest to add a brief discussion about the role of cardiac autophagy during myocardial ischemia and I/R injury. A potential crosstalk between endothelial vs cardiac autophagy should also be highligthed.

*The authors should provide more details throughout the manuscript when the evidence described is obtained in vivo or in vitro, also distinguishing associative from mechanistic studies.

*Page 9. Mst1, the authors should add a brief description of the Hippo Pathway and its role in cardiovascular diseases.

*Conclusion. Translational approach. I suggest to improve this paragraph, about the therapeutic strategies (natural compounds, mTOR inhibitors, AMPK activators, calorie restriction, caloric restriction mimetics) able to improve endothelial autophagy.

*I suggest to add a table or a figure summarizing relevant studies about the role of endothelial autophagy in CVDs

Author Response

Dear reviewer,

Thank you so much for your worthful suggestions.

For your suggestion 1. *Page 4. The authors should also discuss the emerging role of alternative forms of mitophagy in cardiovascular diseases.

Thank your suggestion. We have added a brief discussion about the emerging role of alternative forms of mitophagy in cardiovascular diseases in lines 458-463. Details are listed below:

Of note, mitophagy is pivotal for mitochondrial quality control and plays a dual role in the progression of diverse CVDs. The emerging role of alternative forms of mitophagy in CVDs is well summarized in Pedro, Morales, and Li’s studies [13, 208, 209]. However, current studies are mainly focused on the role of mitophagy in cardiomyocytes, vascular smooth muscle cells and ECs in the aorta, their roles in coronary ECs are less mentioned and further studies are warranted.

For your suggestion 2, add a brief discussion about the role of cardiac autophagy during myocardial ischemia and I/R injury.

Thank your suggestion. We have added a brief discussion about the role of cardiac autophagy and its crosstalk with endothelial autophagy between lines 252-257. Details are listed below:

Importantly, ischemia and I/R injury also evoke dramatical autophagic flux in cardi-omyocytes, which could either serve as a pro-survival mechanism to meet metabolic demands and eliminate damaged cellular components and organelles, or function as a pro-death mechanism to initiate apoptosis. The crosstalk between endothelial and cardiac autophagy is of great interest but poorly understood. Further investigation is urgently needed.

For your suggestions 3 and 6, provide more details about whether the evidence described is obtained in vivo or in vitro, distinguish associative from mechanistic studies and add a summarized table.

Thank your suggestions. We have added a table that summarized all the mechanistic studies about the role of endothelial autophagy in CVDs. Detailed information about the experimental approach (in vivo or in vitro) was listed in the table.

For your suggestion 4, Mst1, the authors should add a brief description of the Hippo Pathway and its role in cardiovascular diseases.

Thank you for your concern. We have added a brief description of the Mst1 Pathway and its role in cardiovascular diseases in lines 386-388. Details are listed below:

Mst1 (mammalian sterile 20-like kinase 1) is a serine/threonine kinase that functions as a negative regulator of autophagy in the heart by enhancing the binding of Beclin1 to Bcl-2 and promotes apoptosis by releasing Bcl-2 from Bax [182, 183].

For your suggestion 5, for the translational approach.

Thank your suggestion. We have added a brief discussion about the translational approach in lines 464-467. Details are listed below:

In addition, autophagy-targeted pharmacological and nutritional interventions, including mTORC1 inhibitors, AMPK activators, caloric restriction, caloric restriction mimetics, natural compounds, and specific miRNAs, are emerging as potential therapeutic candidates in patients with CVDs [9]

Reviewer 2 Report

The manuscript by Zhao et al. entitled "Endothelial Autophagy in Coronary Microvascular Dysfunction and Cardiovascular Disease" presents the close analysis of data covering the results of clinical and experimental studies regarding the role of the coronary microvascular endothelial autophagy in the genesis of cardiovascular disease including heart failure with preserved ejection fraction, coronary artery disease, diabetic cardiomyopathy, and some others; the role of endothelial autophagy in angiogenesis, cardiac fibrosis mediated by endothelialmesenchymal transition, as well as percutaneous coronary intervention are also reviewed. The detailed description and analysis of clinical and experimental material is preceded by a concise characterization of the coronary microvascular network and autophagy in its main variants (macro microautophagy, chaperonemediated autophagy and mitophagy). The material presented is well organized and provides deep insight into the issues addressed in the manuscript.

The reviewer has no major comments.

Minor comments

Line 338: Sucrose is not a nonmetabolized sugar. It metabolizes by sucrase enzyme in tissues. Regarding mannitol, it is poorly absorbed by intestine. Publication numbered as 181 in References should be taken critically.

Authors are recommended to look carefully through the text for misprints.
Some sentences needed editing noticed by reviewer

Lines 172174:
Upon oxidative stress, Wang et al. reported that restrained Bcl2/ BNIP3mediated endstage of autophagymediated the protective role of MicroRNA103 in human cardiac microvascular ECs (CMECs) against H2O2induced oxidative stress...

Lines 182183:
However, according to Sun’s study, the regulator of calcineurin 11L (Rcan11L) overexpressioninducedmitophagy, which contributed to cell survival under hypoxic conditions...

Author Response

Dear reviewer,

Thank you so much for your valuable comments.

For comment 1. Line 338: Sucrose is not a nonmetabolized sugar.

Yes, you are correct.  Generally speaking, sucrose is not a nonmetabolized sugar.

In the publication numbered 181 in references, the authors performed an ex vivo organ culture model. Briefly, intact fetal hearts were isolated and maintained in an artificial organ culture environment, which was maintained in a control medium or supplemented with sucrose or mannitol. They found prolonged exposure of mouse fetal heart to sucrose or mannitol could induce severe lysosomal derangements and prominent autophagy in ECs. In their study, nonmetabolized sugars were narrowly referred to as osmotically active sugars which didn’t readily gain access to the intracellular compartments of heart tissue except by endocytosis. The engulfed sugars were subsequently incorporated into lysosomes, which were absent of sucrase capable of digesting such sugars.

We apologize for the unclear statement. To avoid confusing readers, I deleted the word nonmetabolizable.

For comment 2. Authors are recommended to look carefully through the text for misprints. Some sentences needed editing noticed by reviewer.

Thank you for your comment. Yes, some of the sentences in our manuscript are too complex and hard to read. We apologize for this. We have revised these sentences as follows:

Previous Lines 172‐174 were moved to Lines 192‐195

Upon oxidative stress, Wang et al. reported that MicroRNA-103 could protect human coronary artery ECs (HCAECs) against H2O2-induced injury by preventing Bcl-2/ BNIP3-mediated suppression of end-stage of autophagy.

Previous Lines 182‐183 were moved to lines 205-208

However, according to Sun’s study, the mitophagy induced by Rcan1-1L (regulator of calcineurin 1-1L) overexpression contributed to cell survival under hypoxic conditions.